# Work Shift, Lifestyle Factors, and Subclinical Atherosclerosis in Spanish Male Workers: A Mediation Analysis

**DOI:** 10.3390/nu13041077

**Published:** 2021-03-26

**Authors:** José L. Peñalvo, Elly Mertens, Ainara Muñoz-Cabrejas, Montserrat León-Latre, Estíbaliz Jarauta, Martín Laclaustra, José M. Ordovás, José Antonio Casasnovas, Irina Uzhova, Belén Moreno-Franco

**Affiliations:** 1Non-Communicable Diseases Unit, Department of Public Health, Institute of Tropical Medicine, 2000 Antwerp, Belgium; ellymertens@itg.be; 2Hospital Universitario Miguel Servet, Instituto de Investigación Sanitaria Aragón (IIS Aragón), CIBERCV, 50009 Zaragoza, Spain; ainaramunozc@gmail.com (A.M.-C.); mleon@unizar.es (M.L.-L.); estijarauta@gmail.com (E.J.); martin.laclaustra@unizar.es (M.L.); jacasas@unizar.es (J.A.C.); mbmoreno@unizar.es (B.M.-F.); 3Department of Medicine, Psychiatry and Dermatology, University of Zaragoza, 50009 Zaragoza, Spain; 4Nutritional Genomics and Epigenomics Group, IMDEA Food, CEI UAM + CSIC, 28049 Madrid, Spain; Jose.Ordovas@tufts.edu; 5Nutrition and Genomics Laboratory, JM-USDA Human Nutrition Research Center on Aging at Tufts University, Boston, MA 02111, USA; 6Department of Health and Nutritional Sciences, Institute of Technology Sligo, F91 YW50 Sligo, Ireland; uzhova.irina@itsligo.ie; 7Department of Preventive Medicine and Public Health, University of Zaragoza, 50009 Zaragoza, Spain

**Keywords:** work shift, lifestyle habits, subclinical atherosclerosis, cardiovascular disease

## Abstract

(1) Background: Working night shifts has been associated with altered circadian rhythms, lifestyle habits, and cardiometabolic risks. No information on the potential association of working shift and the presence of atherosclerosis is available. The aim of this study was to quantify the association between different work shifts and the presence of subclinical atherosclerosis objectively measured by imaging. (2) Methods: Analyses were conducted on the baseline data of the Aragon Workers Health Study (AWHS) cohort, including information on 2459 middle-aged men. Categories of shift work included central day shift, rotating morning-evening or morning-evening-night shift, and night shift. The presence of atherosclerotic plaques was assessed by 2D ultrasound in the carotid and femoral vascular territories. Multivariable logistic models and mediation analysis were conducted to characterize and quantify the association between study variables. (3) Results: Participants working night or rotating shifts presented an overall worse cardiometabolic risk profile, as well as more detrimental lifestyle habits. Workers in the most intense (morning-evening-night) rotating shift presented higher odds of subclinical atherosclerosis (odds ratio: 1.6; 95% confidence interval: 1.12 to 2.27) compared to workers in the central shift, independently of the presence of lifestyle and metabolic risk factors. A considerable (21%) proportion of this association was found to be mediated by smoking, indicating that altered sleep-wake cycles have a direct relationship with the early presence of atherosclerotic lesions. (4) Conclusions: Work shifts should be factored in during workers health examinations, and when developing effective workplace wellness programs.

## 1. Introduction

Lifestyle factors, such as unhealthy eating, excess weight, alcohol consumption, smoking, physical inactivity, and increased stress, have been shown to influence the risk of cardiovascular diseases (CVD) [1]. Other lifestyle factors, such as those related to the disruption of circadian rhythms, although less studied, have also been proposed to impact CVD onset [2]. The sleep-wake cycle, for instance, plays a key role in the regulation of a number of daily physiological activities [3], including eating behaviors [4], peripheral tissue metabolism [5], energy expenditure, body weight regulation [6], and glucose and lipid metabolism [7]. Severe disruption of the sleep-wake cycle, when forcing meal schedules varying more than 12 hours, has been found to negatively impact CVD risk markers such as blood pressure, and postprandial glucose and insulin levels [8].

Work-related factors such as night or rotating shifts severely impact the circadian rhythm [9], resulting in physiological, behavioral, and psychosocial consequences, known as circadian stress [10]. Shift work can increase the risk of CVD by the influence of the three inter-related stress factors that conform with circadian stress. Physiological stress may result in several biological mechanisms related to inflammation or lipid and glucose metabolism; behavioral stress may impact diet and associated weight gain, smoking, diet quality and physical activity; and psychosocial stress may be related to decreased work-life balance or poor recovery following work. The misalignment between the physiological circadian cycle and working schedules is linked to higher risks of obesity, diabetes, and CVD, among other risk factors [8,11,12,13,14].

Lifestyle-related CVD risk factors, including high BMI, unbalanced diet, smoking, and physical inactivity, probably increased by the influence of circadian stress, have also been described to occur with relation to shift work [15,16,17,18,19]. Since altered circadian rhythms can alter lipid metabolism regulation, shift work may impact the development of early presence of atherosclerotic lesions, known as subclinical atherosclerosis.

The measure of subclinical atherosclerosis in several body locations, such as carotid, femoral and coronary arteries, allows one to find evidence of vascular disease before it causes symptoms [20]. In concrete, the presence of femoral plaques has been demonstrated as a strong marker of coronary lesions, being the most prevalent subclinical atherosclerotic alteration with the strongest association with traditional CVD risk factors [20,21].

Although the relationship between the type of shift work and a higher risk of CVD has been previously described, to our knowledge, no previous study has examined the association between the type of shift work and lifestyle factors with the presence of atherosclerotic plaque examined in the subclinical stage by imaging techniques. The aim of this study was to examine this possible association and to understand the influence that lifestyle aspects and traditional metabolic CVD risk factors may play in modulating this relationship.

## 2. Materials and Methods

### 2.1. Study Design, Setting, and Participants

The Aragon Worker’s Health Study (AHWS) is a prospective cohort aiming to investigate the determinants of the development and progression of metabolic abnormalities and subclinical atherosclerosis in 5678 apparently healthy middle-aged workers recruited between 2009 and 2012 [22]. From 2011 to 2014, participants aged 39–59 years (95.0% men) underwent subclinical atherosclerosis imaging and an interview with questionnaires on diet, behavior and lifestyle factors. The present study was carried out on baseline data collected from 2702 participants. Because of the small representation in this population, female workers (*n* = 132) were excluded from the analysis, along with participants with missing information on working shift (*n* = 35) or on the presence of atherosclerosis in bilateral carotid and femoral vascular territories (*n* = 76). The final sample included complete data of 2459 participants. The study was approved by the central ethics committee of Aragón (CEICA), and all study participants provided written informed consent.

### 2.2. Definition of Working Shifts

The distribution of working shifts at the car assembly factory are designed to allow for continuous manufacturing, and includes two fixed shifts (central 08:00–16:00, and night 22:00–06:00) and two rotating shifts (morning-evening (M-E) 06:00–14:00 and 14:00–22:00, and morning-evening-night (M-E-N) 06:00–14:00, 14:00–22:00 and 22:00–06:00). Workers on rotating shifts usually change shifts weekly. 

### 2.3. Atherosclerosis Imaging

The technical details of vascular imaging acquisition have been previously described in detail [22]. In brief, the presence of plaque in the carotid and femoral vascular territories, was assessed with a Philips IU22 ultrasound system (Philips Healthcare, Bothell, Washington). Ultrasound images were acquired with linear high-frequency two-dimensional probes (Philips Transducer L9–3; Philips Healthcare) using the Bioimage Study protocol for the carotid arteries [23] and a specifically designed protocol for the femoral arteries [24]. Inspection sweeps were obtained at right and left sides for the carotid (common, internal and bulb) and femoral territories. Plaque was defined as a focal structure that protrudes into the lumen at least 0.5 mm or is >50% thicker than the surrounding intima-media thickness [21,25]. All measurements were analyzed using electrocardiogram gated frames corresponding to end-diastole (R-wave) [26]. For this study, the presence of subclinical atherosclerosis was defined as the detection of at least one plaque in any of the vascular territories examined. 

### 2.4. Sociodemographic, Clinical, and Biochemical Characteristics

Data on sociodemographic variables were self-reported and included age, educational level (elementary school, secondary school, college degree, or other higher education degree), civil status (married, single/divorced, widowed, or other), number of family members, and type of work (manual labor or sedentary).

Clinical and laboratory data were obtained in the annual medical examination performed in the factory and included BMI, waist circumference, blood pressure, medical history, and the current use of medication. Blood and urine were collected after at least 8 h of overnight fasting. Glucose, total cholesterol, high-density lipoprotein cholesterol (HDL-c), and triglyceride concentrations were determined by enzyme analysis using the ILAB 650 analyzer from Instrumentation Laboratory. Insulin was determined using the Access ultrasensitive chemiluminescence immunoassay from Beckman Coulter. Glycated hemoglobin (HbA1c) was determined by cation exchange chromatography on a reverse-phase column using the ADAMS A1c HA-810 analyzer from Arkray Factory. Low-density lipoprotein cholesterol (LDL-c) concentrations were calculated using the Friedewald formula when the triglyceride levels were <400 mg/dL [27]. Hypertension was determined as systolic blood pressure ≥140 mmHg, diastolic blood pressure ≥90 mmHg, or self-reported use of antihypertensive medication [28]. Diabetes was determined as fasting glucose ≥126 mg/dl or self-reported treatment with hypoglycemic medication [28]. Dyslipidemia was defined as having total cholesterol ≥240 mg/dl, LDL-c ≥160 mg/dl, HDL-c <40mg/dl, or self-reported use of lipid-lowering drugs [29]. Abdominal obesity was determined as a waist circumference of ≥102 cm. The estimation of 10-year Framingham risk score (FRS), which predicts the participant’s risk of presenting with a coronary event in the next 10 years, was calculated based on sex, age, total cholesterol, HDL-c, systolic blood pressure, presence of diabetes, and smoking status, classified as FRS ≤ 10 (low risk), 10 < FRS ≤ 20 (intermediate risk), and FRS > 20 (high risk) [29].

### 2.5. Lifestyle-Related CVD Risk Factors

Information on diet was retrieved by trained interviewers using a 136-item semi-quantitative food frequency questionnaire (FFQ) validated in Spain [30]. Dietary quality was assessed by computing the Alternative MEDiterranean (aMED) score, that considers the intake of total vegetables excluding potatoes, total fruit, nuts, legumes, fish, whole grains, MUFA to SFA ratio, alcohol, and red and processed meat [31]. The aMED score ranged from 0 to 9, respectively, indicating the lowest and highest adherence to a Mediterranean dietary pattern. Information on leisure-time physical activity was self-reported using questionnaires validated in Spain [32]. The participants were asked about the average weekly time spent on 17 different types of physical activity, which was later multiplied by its energy expenditure, expressed in metabolic equivalent transfer (MET) units [33], and added using all activities reported to yield a value of estimation of the total level of physical activity per week (MET-h/week). Sedentary time was defined as the average number of sitting times, considering both working and leisure time in a typical working day. Sleep duration was calculated from self-reported hours of sleep during the week. For participants in the rotational shifts, an average sleep duration was calculated from the different types of weeks. Smoking habits were categorized as current smoking if the participant reported having smoked in the last year, former smoking if the participant had smoked at least 50 cigarettes in his lifetime, but not in the last year, and never smoking. Ever smoking (current and former) versus never smoking was studied in the main analysis. Smoking was quantified with a self-reported number of cigarettes per day.

### 2.6. Statistical Analysis

For descriptive purposes, counts, frequencies, means, and standard deviations were used. Differences across working shifts were assessed using Chi-square for categorical variables or ANOVA for continuous variables, and Bonferroni or Fisher post-hoc tests, respectively. To quantify the association between working shift and the presence of subclinical atherosclerosis, we used multivariable logistic regression models. Associations were reported as odds ratios (OR) and corresponding 95% confidence intervals from adjusted models. Progressive model adjustment was informed from descriptive results according to working shift. We fit an age-adjusted model (model 1) and a multivariable (model 2) model additionally including socioeconomic (education level) and lifestyle-related variables (smoking status, sleeping hours (in categories), sitting time (in hours), aMED score, and consumption of coffee) and a final multivariable model (model 3) excluding identified mediators from variables introduced in model 2. The central shift was used as a reference group as it was considered a standard working schedule. All analyses were performed using R version 3.6.2.

### 2.7. Mediation Analysis

A mediation analysis was conducted to explore the potential metabolic risk factors as intermediate mechanisms to explain the association between shift work and the presence of atherosclerosis and confirm whether lifestyle factors act as confounders or mediators in the association. Mediation was investigated using effect decomposition within a counterfactual framework [34]. This framework enables one to decompose the total effect into an indirect effect through the hypothesized intermediate variables (mediators) and the remaining direct effect. Natural direct effects, natural indirect effects and total effects were estimated using the R package ‘medflex’ [35] built upon the class of natural effect models (NEM) based on a counterfactual approach, as introduced by Lange et al. [36] and Vansterlandt et al. [37]. For the counterfactual, we duplicated each observation in the original dataset four times and created an artificial exposure, i.e., work shift*, which is equal to the original work shift for the first replication and equal to the other work shifts for the second, third and fourth replications. In this duplicated dataset, the presence of subclinical atherosclerosis was imputed using multivariate models fitted to the original dataset, as specified in the statistical analysis with the additional inclusion of the mediator under study. Subsequently, the NEM was fitted to the extended data excluding the mediator under study but including the additional variable work shift*. The work shift coefficient represents the natural indirect log-odds ratio, and the coefficient of work shift* represents the natural direct log-odds ratio. Results were expressed as exponentiated coefficients (OR), and 95% and confidence intervals. Because of multiple comparisons in these mediation analyses, a Bonferroni-adjusted *p*-value < 0.01 was considered to be statistically significant.

## 3. Results

Participants in the AWHS cohort are factory workers, mostly middle-aged men (Table 1).

Because of the factory continuous-operation mode, the majority of the participants worked in rotating shifts (81%), with the M-E modality being the most common shift (61%), and the rest being evenly distributed in the fixed shifts central and night (8.5% and 10%, respectively). Workers in the AWHS cohort are characterized by high employee retention with an average time in the company of 33.2 years (Table 1), with slightly higher times among rotating M-E-N and night shift workers compared to central (*p* < 0.001), or rotating M-E (*p* = 0.001).The central shift included the highest proportion of workers with a college degree (42%) and mostly comprised office staff (77%), as this shift represents standard working hours and administrative operations. These figures contrasted significantly with the night and rotating shifts where most of the workers had elementary school education and belonged almost entirely to the manual labor workforce. In line with the nature of the work type, central shift workers reported spending approximately 1 h more of sitting time, although the overall level of physical activity and TV hours did not differ between groups. Sleeping time was generally less among rotating shifts, and particularly among night shift workers. Active or former smokers were more frequent among the night shift workers. Dietary habits were notably different among central shift workers, scoring the highest on aMED and presenting overall lower caloric intakes, and lower consumption of unhealthy items such as processed red meat, *trans* fats, sodium, sugary drinks, and coffee (Appendix A).

The prevalence of CVD risk factors associated with atherosclerosis varied greatly between shifts (Table 2). Although there were no notable differences in the prevalence of hypertension, dyslipidemia or diabetes, there were differences in the mean values of blood pressure and markers of glucose metabolism unfavorable to both rotating and night shifts. Similarly, glycemic markers and plasma lipid fractions seem to be worse overall for the rotating shifts. The most noticeable difference between shifts corresponds to obesity. Abdominal and BMI-based obesity rates are much larger in the rotating M-E-N and night shifts, with approximately 85% of the workers with excess weight. Workers on both rotating and night shifts presented worse Framingham Risk scores (FRS) with larger fractions of people in the higher levels of risk belonging to rotating M-E (33%), rotating M-E-N (35%) or night (40%) staff, when compared with the central (28%) shift.

The prevalence of subclinical atherosclerosis, defined as at least one plaque in either carotids or femoral territories, was high overall among participants (64%). However, it was more frequently observed among night shift workers (69%), rotating M-E-N (66%), and rotating M-E (64%), than central shift workers (56%) (Table 3). In age-adjusted models (model 1), working in any shift other than the central shift was associated with a higher prevalence of plaque with similar odds across the three groups. However, in the multivariable-adjusted model correcting for potential confounders identified in the descriptive tables (Model 2), only working in the rotating M-E-N shift remained associated with higher odds of atherosclerosis (OR 1.46 95%; CI: 1.02, 20.9). Confounding variables were selected based on their association with shift work and their association with atherosclerosis risk in the present study and not considered as an intermediate in the pathway between shift work and atherosclerosis risk. However, the latter might be questioned especially for lifestyle-related factors; therefore, variables thought to potentially be in the causal pathway of the association between shift work and the presence of atherosclerosis were tested in mediation analyses. For the rotating M-E-N shift, smoking was observed to lay in the pathway of the association, explaining almost 23% of the association through indirect effects (Table 4). Plasma glucose was also observed to behave as a mediator, although the effect was small. Once mediators were excluded from the model (Model 3), the estimate of the association between working in the rotating M-E-N and the presence of atherosclerosis increased significantly (OR 1.60, 95%; CI: 1.12, 2.27).

Albeit no association was observed between working in the night shift or the rotating M-E (Table 3), mediation analyses were also carried out to identify any interaction with other variables. No other explanatory associations were observed (Appendix A).

## 4. Discussion

This is the first study to examine the potential association between working shifts and the presence of atherosclerotic plaque in asymptomatic individuals. In this sample of Spanish men workers without prior history of CVD, we observed that an intense rotating shift (M-E-N) was associated with higher odds of atherosclerosis. This association was independent of metabolic risk factors but was explained partly by lifestyle-related variables such as smoking.

Previous analyses by Rizza et al. [14] on a group of 88 rotating-night shift workers, and 35 former-night shift workers, revealed significantly higher levels of inflammatory, cardiometabolic risk markers, and carotid intima media thickness (cIMT), in comparison with a group of 64 day-only workers. In multivariable models, the association between rotating night-shift and cIMT was also significant [14]. Our results are in agreement with the association between rotational shifts and markers of atherosclerosis, even adding evidence of the association with the presence of plaque. Carotid plaque measurement has been shown to improve significantly the risk prediction of major cardiovascular events [38], and the accuracy for the diagnosis of coronary artery disease, when compared with the cIMT measurement [26].

Frequent changes in schedules, and disruption of wake sleep cycles, may also lead to further neurohormonal and metabolic dysregulations such as inflammation, impaired glucose tolerance, and elevated mean arterial blood pressure levels [8,14], which in turn is associated with increased risk of type 2 diabetes, metabolic syndrome, and coronary heart disease [11,13,39]. Metabolic disturbances could be caused by alterations of melatonin, leptin, or cortisol secretion. These molecules are regulated by the circadian system, where their secretion is altered by light in the night, so that a night shift could inhibit or over-stimulate their secretion. They are in charge of several processes, such as sleep/wake rhythm, blood pressure regulation, appetite suppression, speed of metabolism, fat, protein or carbohydrate metabolism, among others [17,40]. Due to the wide variety of processes in which these molecules are involved, it is not surprising that several health problems can be associated with the disruption of these rhythms in humans [12,41].

Our study suggests that a disruption of wake-sleep cycles may also involve altered lifestyle behaviors. Intense rotating shifts involving frequent schedule adjustments may hinder the adherence to healthy habits. In our study for instance, smoking was identified as a mediator in the association. Active or former smokers were more frequent among night shift workers when compared to rotating M-E-N (*p* = 0.004), or central shift (*p* < 0.001) workers. These results are in concordance with previous studies that describe higher odds of being a current smoker among ever night shift workers [18]. The dysregulation of circadian rhythm, which can increase the circadian stress and in turn result in sleep disorders, digestive problems or decreased function and concentration [42], can be the cause of maintained smoking use to release stress. Indeed, fixed overnight work has been described as likely to negatively affect smoking cessation compared to other work schedules [43].

Our study also highlighted the influence of shift work on dietary habits. Diet quality was better among central shift workers compared to rotating M-E (*p* < 0.001), and night shifts (*p* = 0.001). Central shift workers had a higher adherence to a Mediterranean diet and maintained a healthier dietary pattern than rotating and night shifts. Rotating and night shifts had higher energy intakes and greater intakes of *trans* fats, salt, and soft drinks, compared to central shifts. These results are in line with other cross-sectional studies that associated habitual disruption of the circadian system with higher consumption of soft drinks and coffee, and lower consumption of vegetables and whole grains, and lower overall adherence to a healthy diet [18,19,44,45,46,47]. Several studies have shown that work shifts, specifically night shift work, affect eating habits and food choices, leading to unhealthy eating patterns. These suboptimal dietary patterns could be the cause of above 85% of participants in rotating or night shifts having excess weight, resulting in larger abdominal and BMI-based obesity rates, as have been previously described [17,18,48].

In addition to an unhealthier dietary profile, we expected to observe that night and rotating shifts are associated with other unhealthy lifestyle behaviors such as higher alcohol intake and physical inactivity, as was previously described [49,50,51]. However, we did not observe any significant difference in alcohol intake or physical inactivity between working shifts in our study. Low levels of physical activity have been previously reported in shift workers from several countries, suggesting that inactivity among shift workers may not be related to sociocultural backgrounds but associated to work schedules [52]. Recently, Cheng and co-workers described that only shift work without night shifts was associated with physical inactivity among men (OR 1.38, 95%; CI: 1.09–1.74); on the contrary, workers often working night shifts had a lower risk of physical inactivity [15]. Moreover, Loef and co-workers have shown that leisure time and occupational physical activity levels are similar among shift workers and non-shift workers [53]. Notwithstanding, it is widely accepted that demanding work schedules, circadian rhythm disruption and reduced time availability for physical activity may explain the possible physical inactivity associated with shift workers. In our study, and because of the office-type work performed by the central shift workers, we observed a negative association between working shift and sitting time with central workers spending more time sitting than workers in the other shifts *(p* < 0.001 for all). Rotating and night shift workers were more likely to be involved in the manual labor work, which required less sitting time. Sleeping time reported by central shift workers was different *(p* < 0.001) than sleeping time for rotating or night shift, with central workers reporting more hours of sleep than workers in the other shifts (*p* < 0.05 for all). No differences between rotating shifts (*p* = 0.501), or night and rotating M-E (*p* = 0.182) or night and rotating M-E-N (*p* = 0.395) were observed indicating a clear distinction between central and non-central worker’s sleeping time.

The present study examined the mediating role of lifestyle-related factors and biological risk factors in the association between shift work and atherosclerosis risk. The mediating factors were selected based on their association with exposure and outcome and supported by literature suggesting that the circadian disruption due to shift work affects lifestyle and biological risk factors [10]. The lifestyle-related factors were treated as confounding variables in our most adjusted model. However, they should also be considered as a mediator. This can be illustrated by the lifestyle variable smoking status, since having a smoking history is more prevalent among rotating shift workers (confounding) [54], but smoking can also be a way to cope with the feeling of sleepiness and stress (mediator) [55,56,57]. Likewise, shift workers may, due to their cultural background, select different foods than day-time workers (confounding), while the consumption of foods rich in saturated fats and soft drinks was higher when working in shifts (mediator of an unhealthy diet) [58,59,60].

Our study has several strengths and limitations worth mentioning. As far as we are aware, our study is one of the very few assessing the effect of disturbed wake-sleep routines, lifestyle risk factors and the presence of subclinical atherosclerosis. A major strength of our study is the possibility of evaluating its effect in middle-aged asymptomatic populations using an objective measurement of atherosclerosis. An important limitation is its cross-sectional design and inability to completely exclude residual confounding, particularly concerning socioeconomic variables, even though multivariable adjustment was used in the models. Moreover, it cannot be ruled out that, first, there is a possibility that participants with awareness regarding their health status might have modified their lifestyle; second, dietary and physical activity measurement errors could also have taken place. Lastly, the use of pharmacological therapy might influence variables reflecting levels of metabolic risk factors in some participants.

## 5. Conclusions

In conclusion, in our population, distinctively worse lifestyle and cardiometabolic risk factor profiles can be observed for night and rotating work shifts, when compared to central shift workers. In particular, rotating shifts are associated with higher odds of prevalent atherosclerosis. Work shift should be factored in during workers’ health examinations, and when developing effective workplace wellness programs.

## Figures and Tables

**Table 1 nutrients-13-01077-t001:** Demographics, lifestyle- and job-related characteristics of the AWHS participants according to work shift.

	Total (*n =* 2459)	Central (*n* = 207)	Rotating M-E (*n* = 1493)	Rotating M-E-N (*n* = 499)	Night (*n* = 260)	*p*-Value
Age, years	50.9 ± 3.93	51.4 ± 4.15	50.6 ± 3.94	51.0 ± 3.82	51.8 ± 3.62	<0.001
Time in the company, years	33.2 ± 4.67	31.3 ± 8.84	33.0 ± 4.12	34.0 ± 3.95	34.2 ± 33.2	<0.001
Educational level						
Elementary school, *n* (%)	1232 (50)	32 (15)	914 (61)	138 (28)	148 (57)	<0.001
Secondary school, *n* (%)	1073 (44)	88 (43)	537 (36)	343 (69)	105 (40)	
College degree, *n* (%)	134 (5)	87 (42)	30 (2)	13 (3)	4 (2)	
Other, *n* (%)	20 (1)	0 (0)	12 (1)	5 (1)	3 (1)	
Civil status						0.319
Married, *n* (%)	2071 (84)	179 (86)	1235 (83)	429 (86)	228 (88)	
Single/divorced, *n* (%)	351 (14)	25 (12)	236 (16)	62 (12)	28 (11)	
Widower, *n* (%)	17 (1)	0 (0)	11 (1)	4 (1)	2 (1)	
Other, *n* (%)	20 (1)	3 (1)	11 (1)	4 (1)	2 (1)	
Number of family members						0.657
1-2, *n* (%)	518 (21)	44 (21)	309 (21)	115 (23)	50 (19)	
3-4, *n* (%)	1,831 (74)	151 (73)	1,118 (74)	361 (72)	201 (77)	
5 or more, *n* (%)	108 (4)	12 (6)	65 (4)	22 (4)	9 (3)	
Missing, *n* (%)	2 (0.08)	0 (0)	1 (0.07)	1 (0.2)	0 (0)	
Type of work						<0.001
Factory, manual labor; *n* (%)	2165 (88)	48 (23)	1437 (96)	430 (86)	250 (96)	
Office, sedentary, *n* (%)	294 (12)	159 (77)	56 (4)	69 (14)	10 (4)	
Physical activity						
Physical activity, METS-h/week	32.2 ± 22.9	31.5 ± 20.8	32.0 ± 22.8	33.4 ± 23.9	31.4 ± 23.7	0.560
Sitting time, h/day	6.47 ± 1.69	7.50 ± 1.31	6.30 ± 1.72	6.44 ± 1.61	6.63 ± 1.65	<0.001
Missing, *n* (%)	20 (0.8)	2 (1.0)	11 (0.7)	4 (0.8)	3 (1.2)	
TV time, h/day	3.59 ± 2.36	3.35 ± 2.32	3.56 ± 2.35	3.67 ± 2.42	3.84 ± 2.39	0.124
Sleep time						0.010
Less or equal to 5 h/day, *n* (%)	154 (6)	7 (3)	95 (6)	37 (7)	15 (6)	
6 h/day, *n* (%)	845 (34)	52 (25)	508 (34)	182 (36)	103 (40)	
7 h/day, *n* (%)	1112 (45)	116 (56)	679 (45)	218 (44)	99 (38)	
More or equal to 8 h/day, *n* (%)	343 (14)	32 (15)	211 (14)	62 (12)	38 (15)	
Missing, *n* (%)	5 (0.2)	-	-	-	5 (1.9)	
Smoking status						<0.001
Non-smokers, *n* (%)	565 (23)	72 (35)	326 (22)	119 (24)	48 (18)	
Former smokers, *n* (%)	1032 (42)	88 (43)	596 (40)	240 (48)	108 (42)	
Active smokers, *n* (%)	862 (35)	47 (23)	571 (38)	140 (28)	104 (40)	
Diet						
Total energy, Kcal/d	2910 ± 741	2590 ± 617	2960 ± 745	2940 ± 742	2850 ± 741	<0.001
aMED score	4.07 ± 1.74	4.56 ± 1.81	3.93 ±1.73	4.31 ± 1.73	3.95 ±1.67	<0.001
Missing, *n* (%)	5 (0.2)	-	4 (0.3)	-	1 (0.4)	

Data are presented as *n* (%) or mean ± SD; Central—Central-day shift; M-E—morning/evening rotation shift; M-E-N—morning/evening/night rotation shift; Night—night shift; METs—Metabolic Equivalent of Tasks; aMED—alternative Mediterranean diet index; MUFA—mono-unsaturated fatty acids; SFA—saturated fatty acids, *p*-value obtained using ANOVA for continuous variables and Chi-square for categorical variables.

**Table 2 nutrients-13-01077-t002:** Cardiovascular risk factor of AWHS population by working shift.

	Total (*n* = 2459)	Central (*n* = 207)	Rotating M-E (*n* = 1493)	Rotating M-E-N (*n* = 499)	Night (*n* = 260)	*p*-Value
Family history CVD	31 (1)	2 (1)	20 (1)	6 (1)	3 (1)	0.968
Hypertension, *n* (%)	697 (28)	56 (27)	407 (27)	148 (30)	86 (33)	0.228
SBP, mmHg	125 ± 14.1	123 ± 12.5	126 ± 14.3	125 ± 14.2	127 ± 13.8	0.008
DBP, mmHg	82.6 ± 9.47	81.7 ± 9.01	82.6 ± 9.63	83.2 ± 9.29	82.6 ± 9.21	0.282
Diabetes, *n* (%)	145 (6)	10 (5)	87 (6)	32 (6)	16 (6)	0.871
Glucose, mg/dL	98.0 ± 17.7	103 ± 15.2	98.8 ± 16.0	98.1 ± 18.9	89.3 ± 22.6	<0.001
Insulin, uU/mL	7.96 ± 5.89	7.34 ± 5.59	7.93 ± 5.90	8.77 ± 6.17	6.87 ± 4.45	0.010
*Missing, n (%)*	*1101 (44.8)*	*80 (38.6)*	*682 (45.7)*	*203 (40.7)*	*136 (52.3)*	
HbA1c, %	5.56 ± 0.56	5.44 ± 0.57	5.55 ± 0.46	5.61 ± 0.70	5.59 ± 0.75	0.025
Missing, *n* (%)	1003 (40.8)	78 (37.7)	604 (40.5)	189 (37.9)	132 (50.8)	
Dyslipidemia, *n* (%)	404 (16)	40 (19)	236 (16)	73 (15)	55 (21)	0.069
Total cholesterol, mg/dL	220 ± 36.4	216 ± 36.5	220 ± 36.5	219 ± 33.2	225 ± 41.0	0.074
HDL-c, mg/dL	52.9 ± 11.4	53.9 ± 11.5	52.9 ± 11.3	51.9 ± 11.4	54.1 ± 11.8	0.039
LDL-c, mg/dL	137 ± 32.9	135 ± 34.5	136 ± 33.2	136 ± 29.6	140 ± 35.3	0.282
Triglycerides, mg/dL	153 ± 102	135 ± 91.6	153 ± 105	158 ± 92.2	152 ± 107	0.048
Waist circumference (cm)	97.9 ± 9.25	96.8 ± 8.76	97.3 ± 9.17	99.3 ± 9.33	100 ± 9.36	<0.001
Missing, *n* (%)	57 (2.3)	5 (2.4)	36 (2.4)	10 (2.0)	6 (2.3)	
BMI (kg/m^2^)	27.9 ± 3.43	27.2 ± 3.17	27.6 ± 3.42	28.5 ± 3.40	28.4 ± 3.52	<0.001
Missing, *n* (%)	6 (0.2)	-	3 (0.2)	2 (0.4)	1 (0.4)	
BMI categories						<0.001
Normo-weight (BMI < 25), *n* (%)	478 (19)	43 (21)	330 (22)	65 (13)	40 (15)	
Overweight (BMI 25-30), *n* (%)	1400 (57)	130 (63)	849 (57)	275 (55)	146 (56)	
Obese (BMI > 30), *n* (%)	575 (23)	34 (16)	311 (21)	157 (31)	73 (28)	
Missing, *n* (%)	6 (0.2)	-	3 (0.2)	2 (0.4)	1 (0.4)	
Central Obesity, *n* (%)	778 (32)	57 (28)	421 (28)	195 (39)	105 (40)	<0.001
Framingham Risk Score	0.18 ± 0.10	0.16 ± 0.10	0.18 ± 0.10	0.18 ± 0.11	0.19 ± 0.10	0.004
Framingham Risk Score categories						0.011
Low (score < 10), *n* (%)	555 (23)	60 (29)	336 (23)	118 (24)	41 (16)	
Intermediate (score 10-20), *n* (%)	1080 (44)	90 (43)	668 (45)	208 (42)	114 (44)	
High (score > 20), *n* (%)	824 (34)	57 (28)	489 (33)	173 (35)	105 (40)	

Data are presented as *n* (%) or mean ± SD; Central—Central-day shift; M-E—morning/evening rotation shift; M-E-N—morning/evening/night rotation shift; Night—night shift; SBP—systolic blood pressure; DBP—diastolic blood pressure; HbA1c—glycated hemoglobin; BMI—Body mass index; LDL—Low density lipoprotein; HDL—High density lipoprotein; *p*-value obtained using ANOVA for continuous variables and Chi-square for categorical variables.

**Table 3 nutrients-13-01077-t003:** Odds of subclinical atherosclerosis (presence of plaque in the femoral or carotid territories) by work shift in the AWHS population.

	Model ^b^	Central *n* = 207 (%)	Rotating M-E *n* = 1493 (%)	Rotating M-E-N *n* = 499 (%)	Night *n* = 260 (%)
Atherosclerosis, *n* (%) ^a^		117 (56%)	959 (64%)	333 (66%)	180 (69%)
	1	1 (ref)	1.53 ** [1.13,2.07]	1.63 ** [1.16,2.29]	1.68 ** [1.14,2.48]
	2	1 (ref)	1.20 [0.86,1.68]	1.46 * [1.02,2.09]	1.23 [0.81,1.87]
	3	1 (ref)	-	1.60 ** [1.12,2.27]	-

Exponentiated coefficient (Odds Ratios (OR) and 95% confidence intervals in brackets), * *p* < 0.05, ** *p* < 0.01; ^a^ Defined as the presence of at least 1 plaque in either femorals or carotids. ^b^ Progressive model adjustment consisted of Model 1 adjusted for participant’s age; Model 2 further adjusted for socio-economic confounders, i.e., educational level; and lifestyle-related confounders, i.e., smoking status, sitting time, sleep duration, adherence to Mediterranean diet and coffee intake; Model 3 removing the adjustment for smoking, as it was identified as potential mediators in the pathway of rotating M-E-N shift and atherosclerosis (see Table 4, and Appendix A). 2429 observations used (98.8%).

**Table 4 nutrients-13-01077-t004:** Estimation of the mediation effect of lifestyle and biological risk factors in the association between work shift (Morning-Evening-Night versus central shift) and the presence of subclinical atherosclerosis.

Mediator	Model ^a^	Natural Direct Effect ^b^			Natural Indirect Effect ^b^			Total Effect ^b^			% Mediated ^b^
		*OR*	95%CI	*p*-Value	*OR*	95%CI	*p*-Value	*OR*	95%CI	*p*-Value	
Lifestyle risks											
Smoking	2	1.43	[1.01; 2.03]	0.042	1.11	[1.03; 1.20]	0.005	1.60	[1.12; 2.28]	0.010	22.9
Sleep	2	1.46	[1.01; 2.10]	0.042	1.03	[1.00; 1.06]	0.076	1.50	[1.04; 2.15]	0.028	-
Sitting	2	1.46	[1.01; 2.10]	0.042	0.96	[0.90; 1.02]	0.163	1.40	[0.97; 2.01]	0.069	-
Mediterranean diet	2	1.46	[1.01; 2.10]	0.042	1.02	[0.99; 1.05]	0.128	1.49	[1.03; 2.14]	0.032	-
Coffee	2	1.46	[1.01; 2.10]	0.042	1.00	[0.98; 1.02]	0.951	1.46	[1.01; 2.10]	0.043	-
Metabolic risks											
SBP	2	1.39	[0.96; 2.00]	0.078	1.05	[1.00; 1.10]	0.036	1.46	[1.01; 2.11]	0.043	-
	3	1.53	[1.07; 2.18]	0.019	1.05	[1.00; 1.09]	0.039	1.60	[1.12; 2.28]	0.010	9.8
HbA1c	2	1.33	[0.87; 2.05]	0.192	1.06	[0.99; 1.13]	0.087	1.41	[0.92; 2.16]	0.112	-
	3	1.44	[0.94; 2.21]	0.094	1.07	[1.00; 1.15]	0.046	1.55	[1.02; 2.35]	0.042	-
BMI	2	1.45	[1.00; 2.09]	0.049	1.02	[0.98; 1.06]	0.458	1.47	[1.02; 2.11]	0.040	-
	3	1.60	[1.12; 2.29]	0.010	1.00	[0.96; 1.04]	0.956	1.60	[1.13; 2.29]	0.009	-
Waist	2	1.46	[1.01; 2.12]	0.047	1.03	[1.00; 1.06]	0.076	1.50	[1.04; 2.18]	0.032	-
	3	1.62	[1.12; 2.32]	0.010	1.03	[0.99; 1.06]	0.105	1.66	[1.16; 2.38]	0.006	-
HDL cholesterol	2	1.42	[0.99; 2.04]	0.060	1.03	[1.00; 1.06]	0.069	1.46	[1.01; 2.10]	0.042	-
	3	1.53	[1.08; 2.19]	0.018	1.04	[1.00; 1.08]	0.036	1.60	[1.12; 2.28]	0.010	8.5
Triglycerides	2	1.42	[0.99; 2.05]	0.058	1.03	[1.00; 1.05]	0.047	1.46	[1.01; 2.10]	0.042	-
	3	1.54	[1.08; 2.19]	0.018	1.04	[1.01; 1.08]	0.018	1.60	[1.12; 2.28]	0.010	8.5
Framingham Risk Score	2	1.37	[0.95; 1.96]	0.090	1.07	[1.01; 1.13]	0.021	1.46	[1.01; 2.10]	0.042	-
	3	1.38	[0.97; 1.96]	0.070	1.16	[1.07; 1.25]	<0.001	1.60	[1.12; 2.28]	0.010	-

Exponentiated coefficient (Odds Ratios (OR) and 95% confidence intervals in brackets). Because of multiple comparisons, Bonferroni-adjusted *p*-values below or equal to 0.01 for lifestyle risks, and below or equal to 0.003 for metabolic factors, were considered to be statistically significant. ^a^ Model 2 adjusted for age, participant’s educational level, and lifestyle-related confounding variables, except the lifestyle-related variable under study. Model 3 removed the adjustment for smoking, as it was identified as a potential mediator in the pathway of work shift and atherosclerosis. ^b^ Estimations of the natural direct, indirect and total effects for rotating Morning-Evening-Night versus central shift.

## Data Availability

The data presented in this study are available on request from the corresponding author. The data are not public due to ethical reasons.

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
