# Peer review of "Work Shift, Lifestyle Factors, and Subclinical Atherosclerosis in Spanish Male Workers: A Mediation Analysis"

_nutrients, 2021, doi:10.3390/nu13041077_

Round 1

Reviewer 1 Report

The objective of this study is to examine the association between work shifts, life-style, metabolic risk factors, and the presence of subclinical atherosclerosis.  The results reported that rotating morning-evening-night shift workers had higher odds of subclinical atherosclerosis than day shift workers and that 21% of the association was mediated by smoking.  The odds for rotating morning-evening shift and night shift workers were not different from day shift workers.  The authors indicated that the altered sleep-wake cycle has a direct relationship with the early presence of atherosclerotic lesions.  There are several comments that should be addressed.

It should be noted that the odds of subclinical atherosclerosis for rotating morning-evening shift and night shift workers were not different from that for day shift workers in Model 2 (Table 2), which adjusted age, socio-economic cofounders, and life-style cofounders. In contrast, the odds for them were higher than that for day shift workers in Model 1, which adjusted for age (Table 2).  This indicates that it is unclear whether the altered-sleep wake cycle has a direct relationship with the early presence of atherosclerotic lesions.

Materials and Methods

The effect of years of experience of each work shift on the presence of atherosclerotic lesions should be examined.  The participants may change work shifts several times.  Additionally, years of exposure should be considered in the presence of atherosclerotic lesions.

“Sedentary time was defined as the average number of sitting times, considering both working and leisure time in a typical working day.“

It is unclear why sedentary time was assessed only for working days.

“Sleep duration was calculated from self-reported hours of sleep during the week.”

It is unclear which week of sleep duration was assessed.

“For the counterfactual, we duplicated each observation in the original dataset four times and created an artificial exposure…”

Is this a bootstrap sampling procedure?  If so, it seems that the number of imputed data set (n=3) is too small.

Results

It is unclear why total energy intake was not used as a confounder.

Discussion

Regarding the effect of disruption of the sleep-wake cycle on atherosclerotic lesions, physiological mechanisms should be discussed which do not involve the metabolic risk factors used as confounders.

The difference in the effect of work shift on atherosclerotic lesions should be discussed.

“This association was independent of metabolic risk factors but was explained partly by lifestyle-related variables such as smoking.”

In mediation analysis, it seems that metabolic risk factors were not used as covariates.  If so, it is not clear whether the association was independent of metabolic risk factors.

“Aside from being a mediator, our study has shown that active or former smokers were more frequent among night shift workers.”

The results of the Chi-square test and post hoc analysis are needed.

“Diet quality was noteworthy different between the participants of the central shift and rotating and night shifts.”

The result of post hoc analysis is needed.

“However, we did not observe any significant difference in alcohol intake or physical inactivity between working shifts in our study.”

Data for alcohol intake is not shown.

“In our study, and because of the office-type work performed by the central shift workers, we observed a negative association between working shift and sitting time.”

The result of post hoc analysis is needed.

“As reported in our previous research, there was no difference in sleep hours between the different shifts [47].”

The P-value of the Chi-square test for sleep duration is less than 0.05.  The result of post hoc analysis is needed.

Table 3

Metabolic risk factors should be used as confounders.

Table 4

Please show odds ratios instead of beta values.  P values should be corrected because of multiple testing.  Metabolic risk factors should be used as confounders.

Author Response

Reviewer 1

The objective of this study is to examine the association between work shifts, life-style, metabolic risk factors, and the presence of subclinical atherosclerosis.  The results reported that rotating morning-evening-night shift workers had higher odds of subclinical atherosclerosis than day shift workers and that 21% of the association was mediated by smoking.  The odds for rotating morning-evening shift and night shift workers were not different from day shift workers.  The authors indicated that the altered sleep-wake cycle has a direct relationship with the early presence of atherosclerotic lesions.  There are several comments that should be addressed.

It should be noted that the odds of subclinical atherosclerosis for rotating morning-evening shift and night shift workers were not different from that for day shift workers in Model 2 (Table 2), which adjusted age, socio-economic cofounders, and life-style cofounders. In contrast, the odds for them were higher than that for day shift workers in Model 1, which adjusted for age (Table 2).  This indicates that it is unclear whether the altered-sleep wake cycle has a direct relationship with the early presence of atherosclerotic lesions.

[Response] Thank you for this remark. We built 2 different models to provide information on how introducing covariates (confounders) in the model modify the effect estimates (Table 3). By including additional confounders, it can be seen that the association disappears for rotating M-E, and night shift, but hold for rotating M-E-N. This implies that the associations seen in model 1 were confounded by socio-economic and lifestyle factors whereas for rotating M-E-N however, the strength of the association decreased, but remained significant. Rotating M-E-N is the most changing shift introducing the largest variation in the participant’s schedule (changing shift on a weekly basis (line 98), and consequently in the sleep-wake cycle. Even though no differences in the sleep duration between the no-central shifts were observed (information clarified in lines 366-onwards), the frequent change of daily routines among these workers seems to be associated with atherosclerosis according to our models, and in this population.

Materials and Methods

The effect of years of experience of each work shift on the presence of atherosclerotic lesions should be examined.  The participants may change work shifts several times.  Additionally, years of exposure should be considered in the presence of atherosclerotic lesions.

[Response] In the car assembly factory, where the AWHS cohort is based, work shifts are assigned at the beginning when the new hire joins the company. Workers changing work shifts throughout their careers are very rare. Furthermore, AWHS workers are characterized by high staff retention, and extended careers in the company with most of them join the company at an early age and staying an average of 33.2 ± 4.67 years. For this reason, the number of years in the company is highly correlated with the age of the worker in our population, and that disallowed its use in the multivariable models. We have added information on this aspect in table 1 and  204-onwards.

 “Sedentary time was defined as the average number of sitting times, considering both working and leisure time in a typical working day.“ It is unclear why sedentary time was assessed only for working days.

[Response] Because our interest was to understand the association of work shift with the presence of atherosclerosis, we opted to separate working days-habits from weekend-habits, as working days-habits will be highly influenced by the work shift and will allow us to better inform our models.

“Sleep duration was calculated from self-reported hours of sleep during the week.” It is unclear which week of sleep duration was assessed.

[Response] This question asked for the average sleep duration in an average/typical week. For those workers in the rotatory shifts, average sleep duration was calculated from the average/typical weeks depending on the rotation turn. Clarification has been added on line 155-onwards.

“For the counterfactual, we duplicated each observation in the original dataset four times and created an artificial exposure…” Is this a bootstrap sampling procedure?  If so, it seems that the number of imputed data set (n=3) is too small.

[Response] We did not sample from or imputed data to the original dataset. The method used for the assessment of mediation is based on models for the marginal distribution of a counterfactual outcome (marginal structural models). When used in nested counterfactuals, these models enable simultaneous modeling of the natural direct and indirect effect of the exposure X on the outcome Y other than through mediator M. Counterfactuals are built by creating a new dataset by repeating each observation in the original dataset twice and including an additional (artificial) variable X*, which is equal to the original exposure for the first replication and equal to the opposite of the actual exposure for the second replication. Complete information about this method can be found in the reference cited in the manuscript [1]

  1. Lange, T.; Vansteelandt, S.; Bekaert, M., A simple unified approach for estimating natural direct and indirect effects. Am J Epidemiol 2012, 176 (3), 190-5.

Results

It is unclear why total energy intake was not used as a confounder.

[Response] We chose to adjust our models by dietary quality which should account for the whole diet instead of energy that only accounts for caloric intake. Including also energy in the model will lead to over-adjusting because of collinearity. Both results for energy and diet quality (aMED) are presented in table 1, and details on food and nutrient consumption are available in the appendix (Table A1).

Discussion

Regarding the effect of disruption of the sleep-wake cycle on atherosclerotic lesions, physiological mechanisms should be discussed which do not involve the metabolic risk factors used as confounders. The difference in the effect of work shift on atherosclerotic lesions should be discussed.

 [Response] We have rearranged the discussion to add clarity between metabolic, lifestyle and physiological drivers of the relationship between work shift and plaque.

“This association was independent of metabolic risk factors but was explained partly by lifestyle-related variables such as smoking.” In mediation analysis, it seems that metabolic risk factors were not used as covariates.  If so, it is not clear whether the association was independent of metabolic risk factors.

 [Response] Mediation analyses included metabolic risk factors, in all three different work shifts (Table 4, A2, and A3) compared to the central shift.

“Aside from being a mediator, our study has shown that active or former smokers were more frequent among night shift workers.” The results of the Chi-square test and post hoc analysis are needed.

[Response] Information on post-hoc analysis was introduced in lines 163-onwards. Chi-sq results are reported in table 1, and post-hoc test result in line 323-onwards.

“Diet quality was noteworthy different between the participants of the central shift and rotating and night shifts.” The result of post hoc analysis is needed.

[Response] Information on post-hoc analysis was introduced in lines 163-onwards. ANOVA results are reported in table 1, and post-hoc test results in line 332-onwards.

“However, we did not observe any significant difference in alcohol intake or physical inactivity between working shifts in our study.” Data for alcohol intake is not shown.

[Response] Alcohol consumption by work shift is presented in the Appendix (Table A1)

“In our study, and because of the office-type work performed by the central shift workers, we observed a negative association between working shift and sitting time.” The result of post hoc analysis is needed.

[Response] Information on post-hoc analysis was introduced in lines 163-onwards. ANOVA results are reported in table 1, and post-hoc test results in line 363-onwards.

“As reported in our previous research, there was no difference in sleep hours between the different shifts [47].” The P-value of the Chi-square test for sleep duration is less than 0.05.  The result of post hoc analysis is needed.

[Response] Thank you for identifying this error. It has been amended and clarified in lines 366-onwards.

Table 3

Metabolic risk factors should be used as confounders.

 [Response] In the author’s opinion metabolic risk factors cannot be considered confounders as they are unlikely to cause the exposure (work shift). For this reason, we carried out mediation analysis to understand which proportion of the association found between work shift and atherosclerosis could be explained by lifestyle or metabolic risk factors.

Table 4

Please show odds ratios instead of beta values.  P values should be corrected because of multiple testing.  Metabolic risk factors should be used as confounders.

[Response] Beta coefficients have been transformed to odds ratios and their corresponding 95% confidence intervals in tables 4, A2, and A3. P-values are corrected for multiple comparisons, and explanation has been added to the table legends, and to the text in line 195.

Reviewer 2 Report

In this manuscript Dr Penalvo et al. report that rotating and night shift workers present an higher risk of atherosclerosis compared to workers doing different types of shift, independently of cardiovascular and metabolic risk factors. This finding results from a large propective cohort study (AHWS) but it includes only male subjects.

 Although the main finfding is certainly relevant and the study is well conducted, the study suffers from some important flaws that need to be improved

Major concerns:

  • This is not the first report analyzing the association between NSW and subclinical atherosclerosis. Of note, very recently, we published an article on this very subject (Rizza S et al. Carotid intimal medial thickness in rotating night shift is related to IL1β/IL6 axis. Nutr Metab Cardiovasc Dis. 2020. doi: 10.1016/j.numecd.2020.05.028. Epub 2020 Jun 7) that authors need to read, comment and include in the study references list.
  • The presence of an atherosclerosic placque is not a form of subclinical atherosclerosis rather than a manifestation of clinical atherosclerosis since it may lead to vascular murmurs as well as intermittent atherosclerosis or stroke. Subclinical atherosclerosis are increased Intimal Medial Tickness or Pulse Wave Velocity (increased arterial stifness). Please edit the text troughout the manuscript following this importan issue.
  • Table 2 and results: it is misleading to show in tables and comment in the text the systolic and diastolic bolood pressure values, lipids as well as fasting glucose levels without information regarding the pharmacological therapy. In fact, it is obvious that some types of drugs may influence the atherosclerotic process (i.e. statins, ace-inhibitors, SGTL-2 inhibitors or GLP-1 agonists). Therefore I suggest to classify the cardiovascular risk factors as yes/no and use them as qualitative variables for analysis.

Minor concers:

To better describe the metabolic disturbances frequently detectable in night shift workers please include and comment the following references: 1) Rizza S et al Night Shift Working Is Associated With an Increased Risk of Thyroid Nodules. J Occup Environ Med. 2020 doi: 10.1097/JOM.0000000000001711, 2) Rizza S et al Monthly fluctuations in 25-hydroxy-vitamin D levels in day and rotating night shift hospital workers. J Endocrinol Invest. 2020. doi: 10.1007/s40618-020-01265-x, 3) Zoto E et al. Effect of night shift work on risk of diabetes in healthy nurses in Albania. Acta Diabetol. 2019 doi: 10.1007/s00592-019-01307-8.

Author Response

REVIEWER 2

In this manuscript Dr Penalvo et al. report that rotating and night shift workers present an higher risk of atherosclerosis compared to workers doing different types of shift, independently of cardiovascular and metabolic risk factors. This finding results from a large propective cohort study (AHWS) but it includes only male subjects. Although the main finfding is certainly relevant and the study is well conducted, the study suffers from some important flaws that need to be improved

Major concerns:

  • This is not the first report analyzing the association between NSW and subclinical atherosclerosis. Of note, very recently, we published an article on this very subject (Rizza S et al. Carotid intimal medial thickness in rotating night shift is related to IL1β/IL6 axis. Nutr Metab Cardiovasc Dis. 2020. doi: 10.1016/j.numecd.2020.05.028. Epub 2020 Jun 7) that authors need to read, comment and include in the study references list.

[Response] Thank you for this remark. We have acknowledged this previous paper in the revised manuscript (lines 294 onwards)

  • The presence of an atherosclerosic placque is not a form of subclinical atherosclerosis rather than a manifestation of clinical atherosclerosis since it may lead to vascular murmurs as well as intermittent atherosclerosis or stroke. Subclinical atherosclerosis are increased Intimal Medial Tickness or Pulse Wave Velocity (increased arterial stifness). Please edit the text troughout the manuscript following this importan issue.

[Response]  Thank you for the comment. In our understanding, carotid plaque and cIMT measurements are the most used early predictors of subclinical atherosclerosis. However, although cIMT is a common option for its detection, recent research carried out in 6102 asymptomatic persons has concluded that cIMT measure did not improve the risk prediction of major cardiovascular events significantly when comparing with carotid plaque measure [1]. Likewise, carotid plaque measurement has shown to be more accurate than cIMT for the diagnosis of coronary artery disease [2]. We have added a clarification on line 294-onwards. Also, participants in our study are asymptomatic individuals, and history of CVD is an exclusion criterium for entering the cohort. This is why we have used the presence of plaque to define subclinical atherosclerosis in our AWHS protocols and previous publications.

  • Table 2 and results: it is misleading to show in tables and comment in the text the systolic and diastolic bolood pressure values, lipids as well as fasting glucose levels without information regarding the pharmacological therapy. In fact, it is obvious that some types of drugs may influence the atherosclerotic process (i.e. statins, ace-inhibitors, SGTL-2 inhibitors or GLP-1 agonists). Therefore I suggest to classify the cardiovascular risk factors as yes/no and use them as qualitative variables for analysis.

[Response] Thank you for this important remark. Unfortunately, we do not have information on medication on our participants, and indeed normal levels of metabolic markers may reflect the effect of drug treatment. In table 2, we also present the information on the prevalence of hypertension, diabetes, and dyslipidemia which is recorded as “self-reported treatment for X”. The prevalence does not differ across shifts, and this has been mentioned in the manuscript as a qualitative assessment (line 228). Furthermore, we have added your comment in the limitation section (line 396).

Minor concers:

To better describe the metabolic disturbances frequently detectable in night shift workers please include and comment the following references: 1) Rizza S et al Night Shift Working Is Associated With an Increased Risk of Thyroid Nodules. J Occup Environ Med. 2020 doi: 10.1097/JOM.0000000000001711, 2) Rizza S et al Monthly fluctuations in 25-hydroxy-vitamin D levels in day and rotating night shift hospital workers. J Endocrinol Invest. 2020. doi: 10.1007/s40618-020-01265-x, 3) Zoto E et al. Effect of night shift work on risk of diabetes in healthy nurses in Albania. Acta Diabetol. 2019 doi: 10.1007/s00592-019-01307-8.

[Response] Thank you for providing these references. They have been added and discussed in the revised manuscript (lines 59, 294-onwards).

References used:

  1. Sillesen, H.; Sartori, S.;  Sandholt, B.;  Baber, U.;  Mehran, R.; Fuster, V., Carotid plaque thickness and carotid plaque burden predict future cardiovascular events in asymptomatic adult Americans. Eur Heart J Cardiovasc Imaging 2018, 19 (9), 1042-1050.
  2. Inaba, Y.; Chen, J. A.; Bergmann, S. R., Carotid plaque, compared with carotid intima-media thickness, more accurately predicts coronary artery disease events: a meta-analysis. Atherosclerosis 2012, 220 (1), 128-33.

Round 2

Reviewer 2 Report

I thank the authors for addressing my comments. I have no more suggestion to propose.